

# Dynamical drivers of free-tropospheric ozone increases over equatorial Southeast Asia

Ryan M. Stauffer[1], Anne M. Thompson[1,2], Debra E. Kollonige[1,3], Ninong Komala[4], Habib Khirzin Al-Ghazali[4], Dian Yudha Risdianto[4], Ambun Dindang[5], Ahmad Fairudz bin Jamaluddin[5], Mohan Kumar Sammathuria[5], Norazura Binti Zakaria[5], Bryan J. Johnson[6], Patrick D. Cullis[6,7]

[1]NASA/GSFC Atmospheric Chemistry and Dynamics Laboratory, Greenbelt, MD, USA
[2]University of Maryland Baltimore County, Baltimore, MD, USA
[3]Science Systems and Applications, Inc., Lanham, MD, USA
[4]National Research and Innovation Agency (BRIN), Jakarta, Indonesia
[5]Malaysian Meteorological Department, Ministry of Natural Resources, Environment and Climate Change (NRECC), Petaling Jaya, Malaysia
[6]Global Monitoring Laboratory, NOAA Earth System Research Laboratory, Boulder, CO, USA
[7]Cooperative Institute for Research in Environmental Sciences, University of Colorado Boulder, Boulder, CO, USA

*Correspondence to*: Ryan M. Stauffer (ryan.m.stauffer@nasa.gov)

## Abstract

Positive trends in tropical free-tropospheric (FT) ozone are frequently ascribed to emissions growth, but less is known about the effects of changing dynamics. Extending a prior study (Thompson et al., 2021; https://doi.org/10.1029/2021JD034691; "T21"), we re-examine Southern Hemisphere Additional Ozonesondes (SHADOZ) ozone trends over equatorial Southeast Asia (ESEA), one of Earth's most convectively active regions, using 25 years (1998-2022) of ozone soundings. T21 posited that early-year positive FT ozone trends at equatorial SHADOZ stations are related to decreasing convection. The 25-year analysis of Kuala Lumpur and Watukosek SHADOZ records finds that +5 to +15% (+2 to +6 nmol mol$^{-1}$) per decade FT ozone trends from ~February-April coincide with large increases in satellite infrared brightness temperatures and outgoing longwave radiation, indicators of declining convective activity. MERRA-2 reanalyses exhibit decreases in upper tropospheric velocity potential and precipitable water, also indicating diminished convection. In contrast, trends in ozone and convective indicators are generally weak the rest of the year. These results suggest that decreases in convective intensity and frequency are primary drivers of FT ozone build-up over ESEA early in the year, i.e., waning convection suppresses lofting and dilution of ozone. Decreasing convection promotes accumulation of biomass burning emissions typical of boreal spring even though satellite FT carbon monoxide trends (2002-2022) over ESEA follow a global decrease pattern. Finally, our results demonstrate the advantages of monthly or seasonally resolved trends over annual means for robust attribution of observed ozone trends, challenging models to reproduce these detailed features in simulations of the past 25 years.



## 1 Introduction

Tropospheric ozone is a vital trace gas that influences climate, vegetation, and human health. It is a greenhouse gas, particularly in the upper-troposphere (**Lacis et al., 1990; de F. Forster and Shine, 1997; Skeie et al., 2020**), and, because it is a strong oxidant, at even modest levels near the surface is detrimental to crop productivity (**Mills et al., 2018** and references therein) and human health (**Fleming et al., 2018a** and references therein). There has recently been an increase in effort to quantify current and historical tropospheric ozone levels (**Gaudel et al. 2018**; **Tarasick et al., 2019**), reduce uncertainties in calculated

trends, characterize its negative consequences for health, and understand its likely future evolution (**Archibald et al., 2020**). Many of these activities have been coordinated under the auspices of the International Global Atmospheric Chemistry (IGAC) community-sponsored Tropospheric Ozone Assessment Report (TOAR; https://igacproject.org/activities/TOAR/TOAR-I) and the ongoing TOAR-II (https://igacproject.org/activities/TOAR/TOAR-II). Detailed examination of the historically observation-sparse tropics is a primary focus of TOAR-II, as routine in-situ ozone profiles were extremely limited until the

inception of the Southern Hemisphere Additional Ozonesondes (SHADOZ; **Thompson et al., 2003**) network in 1998.

Recent studies indicate that equatorial SE Asia (ESEA) is a region of rapid tropospheric ozone growth. An analysis of In-Service Aircraft for a Global Observing System (IAGOS) ozone profiles by **Gaudel et al., (2020)** showed 50th percentile tropospheric ozone trends of +3 to +11 nmol mol$^{-1}$ per decade over Malaysia and +5 to +7 nmol mol$^{-1}$ per decade over Southeast Asia for 1994-2016. Tropospheric column ozone trends derived from satellite products such as OMI/MLS also reported +2 to

+3 Dobson Unit (DU) per decade increases in the same region from 2005-2016 (**Ziemke et al., 2019**). Typical explanations for these rapid changes are the increase in global methane (**Staniaszek et al., 2022**) and a shift of ozone precursor emissions toward tropical latitudes (**Zhang et al., 2016**) that cause growth in background tropospheric ozone amounts. However, the possible effects of dynamics and climate change have been given little consideration. This is somewhat surprising because the interaction of climate oscillations like El Niño-Southern Oscillation (ENSO; **Ziemke et al., 1997; 2015; Lee et al., 2010;**

**Thompson et al., 2011**), Madden-Julian Oscillation (MJO; **Stauffer et al., 2018**) and the Indian Ocean Dipole (**Thompson et al., 2001**) with tropical tropospheric ozone variability is well-established. Even during the intense 1997-1998 ENSO, a large increase in ESEA tropospheric ozone associated with anomalous dynamics preceded the fire pollution impacts (**Thompson et al., 2001**).

One reason for the neglect of climate-dynamic impacts on tropospheric ozone trends may be the tendency to report using a

single (annually averaged) value over some period of interest. Seasonally or monthly resolved analyses are less common (e.g., **Chang et al., 2023;** Section 3.4) but not having seasonal trends may obscure important mechanistic information. In the tropics both pollution sources, e.g., fires, and meteorological factors, e.g., precipitation, convection, are important causes of seasonal and interannual variability in tropospheric ozone (**Thompson et al., 2012; Oman et al., 2013**). Thus, it becomes difficult or impossible to attribute ozone trends to factors that may be changing only in certain months of the year.



Monthly free-tropospheric (FT) to lowermost stratosphere (LMS) ozone profile trends for five tropical SHADOZ ozonesonde stations (three of them combined records), including ESEA, were examined in detail in **Thompson et al., (2021;** from hereon "T21"). The largest 1998-2019 FT ozone trends were found over the combined Kuala Lumpur, Malaysia (KL) and Watukosek, Java, Indonesia (Java) stations, with up to +20% per decade (+0.4 to +0.5 DU per decade) trends in the 10-15 km altitude layer in March and April. While the tropical SHADOZ ozone trends varied seasonally and regionally, consistently large FT ozone

increases were found at all stations in the first half of the year. Using a convective proxy (**Thompson et al., 2011**), T21 demonstrated that the early months of the year are generally convectively active with low tropospheric ozone, when the lofting of air redistributes lower-concentration, near-surface ozone throughout the atmospheric column. It is this mechanism that causes the prototypical "S-shape" in tropical tropospheric ozone profiles, with low near-surface and upper tropospheric ozone. Although the lowest concentration ozone profiles may occur at any time of year, they dominate in the early part of the year at

most SHADOZ sites. In contrast, a mid-troposphere ozone maximum, most prominent in July to November, typically results from the long-range transport of biomass burning emissions (**Thompson et al., 1997; Oltmans et al., 2001; Jensen et al., 2012**). The T21 results showed that the convective proxy had decreased in magnitude during these months over the 22-year analysis, leading the authors to hypothesize that a decrease in convection is causing FT ozone to accumulate, culminating in the large positive ozone trends.

The T21 convective proxy, based on inferring a convective signature in each SHADOZ ozone and potential temperature profile pair, is indirect and difficult to investigate more generally. Therefore, in this study, we examine other proxies for tropical convective and pollution activity, including satellite-based observations, as well as several parameters from a meteorological reanalysis. We begin by updating the T21 SHADOZ ozone trends (22 years) to cover the 25-year period from 1998-2022. We focus on the two ESEA stations from T21, Kuala Lumpur, Malaysia (KL; 2.73°, 101.27°) and Watukosek, Java, Indonesia

(Java; −7.5°, 112.6°), to demonstrate that a decrease in convection is responsible for the large, early-year FT ozone increases. We choose these stations because ESEA has one of the largest tropospheric ozone increases globally (**Gaudel et al., 2018; 2020; Ziemke et al., 2019**), as well as the strongest/most intense convective activity in the tropics. Section 2 presents an overview of the observational and reanalysis datasets examined. Section 3 describes the relationships among ESEA tropospheric ozone and meteorological variability and includes the results of our updated FT ozone and convective trends

calculations. Section 4 is a concluding summary and recommendations for future analysis.

## 2 Datasets and Methods

### 2.1 SHADOZ Ozonesondes

We examine tropospheric ozone variability and trends over 1998-2022 from 5-15 km altitude for two combined SHADOZ ozonesonde station records in the ESEA region: Kuala Lumpur, Malaysia (KL) and Watukosek, Java, Indonesia (Java). The

SHADOZ profiles are the reprocessed/homogenized V06 data described by **Witte et al., (2017)**. The two stations are shown



on **Figure 1** with fire hotspots derived from the Moderate Resolution Imaging Spectroradiometer (MODIS) instruments on the Aqua and Terra satellites, which illustrates typical locations of fire activity that strongly modulates tropospheric ozone in the region. **Figure 1** also shows three highlighted regions: 5° x 5° latitude/longitude squares centered around KL (yellow) and Java (cyan), and a larger area (-12.5° to 12.5° latitude; 90° to 135° longitude), all for which monthly trends of various datasets will later be explored in detail.

A total of 870 SHADOZ V06 (see *Data availability* for links to all datasets used here) ozonesonde profiles are available for KL and Java from 1998-2022. To increase the total sample size for trends calculations, the ozone profiles from the two stations are merged exactly as in T21. Monthly ozone profile climatologies with 100 m vertical resolution for each station are calculated for 1998-2022, and every individual 100 m profile is recomputed as an anomaly from its respective monthly climatology. The ozone anomaly profiles for both stations from 1998-2022 are averaged together to form a monthly ozone anomaly time series of profiles which are then analyzed for trends. This approach serves to both fill temporal gaps from the individual stations and avoid any step changes in ozone during those periods due to slightly differing ozone climatologies at KL and Java.

## 2.2 Satellite Datasets

### 2.2.1 GridSat-B1

We use 11 μm brightness temperatures ($T_b$) measured from a series of geostationary orbiting meteorological satellites at 0.07° x 0.07° spatial and 3-hourly temporal resolution during our study period of 1998-2022 (data are available since 1980; **Knapp et al., 2011**). An image showing the timeline of satellites used in GridSat-B1 and their geographical coverage can be found at https://www.ncdc.noaa.gov/gridsat/images/isccp_coverage_VZA60_nolegend.png. There is complete geographical coverage for our region in **Figure 1** during our study period. The 11 μm channel is the Climate Data Record (CDR) quality infrared window which essentially measures the $T_b$ of cloud tops or the surface in clear skies. Because $T_b$ is closely associated with the intensity and frequency of convection, we use GridSat-B1 to link variations and trends in tropospheric ozone with changes in convective activity.

### 2.2.2 Atmospheric Infrared Sounder Carbon Monoxide

The Atmospheric Infrared Sounder (AIRS; **Aumann et al., 2003**) was launched on the Aqua satellite in May 2002 and measures several meteorological variables and atmospheric species including carbon monoxide (CO). As shown in our previous work (**Stauffer et al., 2016; 2017; 2018**), there is strong correspondence among tropospheric ozone profile and CO anomalies, particularly over the KL and Java stations examined here (cf **Stauffer et al., 2018** Figures 10 and 11). We use the ascending orbit (~1330 local time) daily L3 AIRS CO mixing ratio data at the 500 hPa level to further quantify the relationship between tropospheric ozone and CO, and trends in the two constituents. The AIRS CO data are available from September 2002-December 2022 for our study.





### 2.2.3 NOAA CDR OLR

NOAA's CDR Outgoing Longwave Radiation (OLR; **Lee and NOAA CDR Program, 2018**) product is analogous to the GridSat-B1 $T_b$ in that colder cloud tops (low $T_b$) occur simultaneous with and cause low OLR amounts, and vice versa. We use the monthly, 2.5° x 2.5° resolution dataset to bolster our analysis of $T_b$ trends. The OLR data have global coverage since

1979 and are generated from the High Resolution Infrared Radiation Sounder (HIRS) series of instruments on various polar-orbiting satellites since Television Infrared Observation Satellite – Next Generation to the contemporary Joint Polar Satellite System instruments and MetOp.

### 2.3 MERRA-2 Reanalysis

NASA's Global Modeling and Assimilation Office Modern-Era Retrospective Analysis for Research and Applications, version

2 (MERRA-2; **Gelaro et al., 2017**) contains a comprehensive suite of meteorological output that provides context for the SHADOZ ozonesonde profile variability and trends. We use two collections from MERRA-2: The 3-hourly, instantaneous, pressure-level, assimilated meteorological fields (M2I3NPASM), and the monthly mean, time-averaged, single-level diagnostics (M2TMNXSLV). The daily, 18 UTC U and V winds at 200 hPa from M2I3NPASM were used to calculate velocity potential (VP200). Anomalies of VP200 quantify the magnitude of upper-tropospheric divergence and are a standard metric

for diagnosing large-scale convective activity and the progression of the MJO through the tropics. Negative VP200 anomalies are associated with enhanced convective activity, and vice versa. Monthly averaged precipitable water (PWAT), a measure of available tropospheric moisture and an additional indicator of convective activity comes from the M2TMNXSLV collection. GridSat-B1 ($T_b$), NOAA OLR, and MERRA-2 (VP200 and PWAT) are used collectively to strengthen our results showing the relationship between tropospheric ozone and convection, and their trends over ESEA since 1998.

### 2.3 Self-Organizing Maps applied to Ozonesonde Profiles

Self-Organizing Maps (SOM; **Kohonen, 1995**) are a form of unsupervised machine learning that we employ to generate clusters of similarly "shaped" tropospheric ozone profiles from KL and Java. Specifically, when applied to the 5-15 km ozone mixing ratio profiles as is done here, SOM distinguishes groups of low and high tropospheric ozone at the two SHADOZ stations. SOM applications and results for SHADOZ and the global ozonesonde network are discussed at length in our previous

work (**Stauffer et al., 2016; 2017; 2018**), so only a cursory introduction is given here. We follow the methods of **Stauffer et al., (2018**; cf Section 2.4) on choice of SOM parameters. Specifically, a 2x2 SOM is initialized separately for KL and Java with the 100 m average ozone mixing ratio profiles from 5-15 km as input, and with 1000 iterations of the SOM algorithm, four similarly shaped clusters, or nodes, of ozonesonde profiles are generated. No information other than the ozone mixing ratio profiles and altitude are used as SOM data inputs. We select the first and fourth nodes in our analysis, which contain the

lowest and highest tropospheric ozone amounts, respectively, and demonstrate the corresponding variability in tropospheric





ozone and meteorology (T$_b$ and VP200) and composition (AIRS CO). The SOM analyses guide our interpretation of the relationship between ozone and convective trends in Section 3.2.

## 2.4 Multiple Linear Regression Model

We follow the approach of T21 by applying the NASA/GSFC Multiple Linear Regression (MLR) model to compute 1998-
2022 (2002-2022 in the case of AIRS 500 hPa CO) trends. As described above, the KL and Java ozonesonde records are merged by first calculating ozone mixing ratio anomalies from respective monthly means at each station for the individual profiles. The ozone mixing ratio anomaly profiles at the two stations are then averaged together to generate a single monthly timeseries for KL-Java. This mitigates any potential step-changes in time series should only one station be available in a given time period. The MLR model is described as follows:

$O_3(t) = A(t) + B(t) + C(t)MEI(t) + D(t)DMI(t) + \in (t) ,$                              (1)

In Equation (1), t is month, and coefficients A through D include a constant and periodic seasonal and subseasonal elements. A is the MLR-modeled mean monthly cycle and B is the computed linear trend (shown in the results here). Coefficients C and D correspond to the Multivariate ENSO Index, v2 (MEI.v2; https://www.psl.noaa.gov/enso/mei/) and the Dipole Moment Index (DMI; https://psl.noaa.gov/gcos_wgsp/Timeseries/Data/dmi.had.long.data) which describes the strength of the Indian
Ocean Dipole. The error, or residual term, is given by $\in$(t). Trends are computed for the 100 m vertically averaged KL-Java monthly ozone mixing ratio anomalies, 1°x1° averaged monthly GridSat-B1 T$_b$, VP200, PWAT, and AIRS 500 hPa CO, and 2.5°x2.5° averaged monthly NOAA OLR. The MLR model is run identically for all trends output presented here. Confidence intervals for the MLR model terms are determined with a moving-block bootstrap technique with 1000 resamples to account for autocorrelation in the time series (**Wilks, 1997**). Because our focus is on the 5-15 km ozone trends, we do not include the
Quasi-Biennial Oscillation (QBO) in the MLR as in T21, as effects on ozone from the QBO below ~17 km altitude are negligible. Otherwise, the approach described above is identical to T21 for the KL-Java combined record.

## 3 Results

### 3.1 Demonstrating Geophysical Variability with SOM

The SOM results are presented first to motivate our study of the relationship among tropospheric ozone and convective trends.
The left half of **Figure 2** shows the 5-15 km ozone mixing ratio SOM output for nodes 1 and 4 at KL and Java, with the right half indicating the frequency of ozonesonde launch month corresponding to the member profiles for each SOM node/cluster. The mean ozone mixing ratios (solid black lines) for node 1 closely follow the 20$^{th}$ percentile ozone (cyan dashed line; ~20 nmol mol$^{-1}$) for the entire set of profiles from each station, with cluster membership of ~40% of all profiles. The tropical "S-shape" in ozone is easily distinguishable at Watukosek (**Fig. 2e**) compared to the nearly constant mixing ratios from 5-15 km



at Kuala Lumpur (**Fig. 2a**). Similar profile features occur at Watukosek even with the high ozone cluster 4 (**Fig. 2f**) with, on average, decreasing ozone above 6 km, whereas node 4 Kuala Lumpur ozone increases with height throughout the troposphere (**Fig. 2b**). Both Kuala Lumpur and Watukosek node 4 ozone is approximately double that (~50 nmol mol$^{-1}$) of node 1. Cluster membership for node 4 is approximately 15% for both stations. The low cluster membership for node 4 is logical given that the mean profiles greatly exceed the 80$^{th}$ percentile ozone at nearly all altitudes in the mid-to-upper troposphere at the two

stations. The right side of **Figure 2** indicates how stations with a relatively, e.g., compared to tropical Atlantic and mid-high latitude sites, narrow seasonal ozone cycle benefit from SOM analysis without the constraints of a climatology. The monthly evolution of tropospheric ozone based on the SOM output shows large variability, but node 1 profiles (**Fig. 2c, g**) indicate a double peak in low tropospheric ozone amounts in ~February and ~August, while higher ozone amounts (**Fig. 2d, h**) show peaks in ~May and ~October (with Watukosek showing an isolated peak in January).

The composite anomaly (from 1991-2020 daily climatology) of GridSat-B1 11μm $T_b$ corresponding to the dates and times of the SOM member profiles are shown on **Figure 3**. Node 1 (**Fig 3a, b**) profiles are associated with $T_b$ approximately -4 to -8 K below average and therefore more frequent and/or colder convection. The opposite is true for node 4 (**Fig 3c, d**) profiles, which align with much warmer than average $T_b$ of +6 to greater than +8 K indicating reduced/less frequent convection. This illustrates the effect of large-scale enhanced lofting of near-surface, ozone poor air via convection for generating low tropospheric ozone

amounts (**Fig 3a, b**) or the anomalous suppression of that lofting leading to an accumulation of tropospheric ozone (**Fig 3c, d**).

Similar composite maps corresponding to the dates of the SOM cluster profiles were generated for daily 18 UTC MERRA-2 VP200 anomalies in **Figure 4**. The VP200 maps show that the large-scale upper-tropospheric meteorological conditions associated with low and high tropospheric ozone are clearly distinguishable. The node 1 clusters (**Fig 4a, b**) occur on days

where upper-level divergence is greater than normal (negative VP200 anomalies), which leads to the enhanced convection and the colder cloud tops indicated by $T_b$ on **Figure 3**, and thus low tropospheric ozone. Much like the $T_b$ analysis, the complete opposite is true for node 4 profiles (**Fig 4c, d**). Upper-level divergence is reduced (positive VP200 anomalies), creating unfavorable large-scale conditions for convection, and causing a build-up of tropospheric ozone. Regardless of whether the profile data and ozone clusters used to generate the composite $T_b$ and VP200 anomalies are from Kuala Lumpur or Watukosek,

the spatial patterns are remarkably similar, showing the strong relationship between convection and tropospheric ozone variability over this region of ESEA as a whole.

Because convection, and therefore rainfall, and biomass/agricultural burning also co-vary in ESEA, we investigate the 500 hPa AIRS CO data to link convection, fires (via CO as a proxy for emissions), and tropospheric ozone amounts. **Figure 5** follows the analyses of **Figures 3** and **4** by showing composite anomaly (from monthly 2002/3-2022 climatology) daily Level

3 500 hPa AIRS CO data. As expected, the high ozone clusters at both stations are associated with positive mid-tropospheric CO anomalies of +5 to +10 nmol mol$^{-1}$ (**Fig 5c, d**) found throughout the ESEA region. The lack of convection (**Figs 3** and **4**)





allows a build-up of tropospheric CO in addition to ozone, likely preceded by and coincident with periods of enhanced biomass burning and accumulation of pollution. The AIRS CO corresponding to the low ozone cluster 1 shows differing patterns between the two stations. Tropospheric ozone over Kuala Lumpur is evidently more closely linked to anomalously low CO amounts, with roughly -6 to -8 nmol mol$^{-1}$ anomalies near the station. The low ozone cluster 1 profiles at Watukosek are associated with near-average CO amounts, indicating that low CO over Watukosek is not necessarily a reliable predictor of low tropospheric ozone amounts.

**Figures 2-5** all fit the conceptual model for tropospheric ozone variability in this region of the tropics. In addition to seasonal changes in convection, transport, and biomass burning, anomalous conditions often associated with ENSO and the MJO cause large perturbations to the ozone profile. This conceptual model was visualized here by linking the ozonesonde SOM output to composites of reduced and enhanced convection (11 µm $T_b$ and VP200) and reduced and enhanced tropospheric pollution resulting from the convective conditions (AIRS 500 hPa CO), which cause either a "cleaning out" (enhanced convection) or accumulation of pollution (reduced convection) in the troposphere. Now that the convection/tropospheric pollution links have been demonstrated via the SOM analysis, we follow with MLR trend analyses to determine how changes in $T_b$, VP200, and AIRS CO correspond the seasonal trends in tropospheric ozone at the KL-Java stations.

### 3.2 Trends in FT Ozone and Ancillary Datasets

We present the updated (from T21) MLR monthly ozone trends for the combined KL-Java SHADOZ sites in **Figure 6**, both in nmol mol$^{-1}$ (**Fig 6a**) and percent per decade (**Fig 6b**) for 1998-2022. We focus on the 5-15 km region as with our previous SOM analyses, but **Figure 6a** includes the near-surface to enable comparisons with other recent studies (**e.g., Gaudel et al., 2020**). The near-surface shows positive trends of +4 nmol mol$^{-1}$ per decade or greater in every month except December, maximizing at over +8 nmol mol$^{-1}$ per decade in March and April below 1 km. The vertical pattern of trends in **Figure 6a** agrees with **Gaudel et al., (2020)** who showed the largest IAGOS ozone trends over Malaysia were located below 700 hPa, with more modest trends above that level. For our altitudes of interest from 5-15 km (shown by the black dashed box on **Figure 6a**), the largest positive FT ozone trends, often exceeding the 95% confidence interval shown by the cyan hatching, are found in the months of February-April, with a maximum in March and April near 15 km of over +4 nmol mol$^{-1}$ per decade. The percentage trends in **Figure 6b** show that the mixing ratio trends equate to about +10 to +15% per decade. Trends are weak throughout the rest of the year, generally within ±5% per decade and rarely exceeding the 95% confidence interval. We hypothesize that the large, early year, positive FT ozone trends can be linked to trends in the datasets shown in previous figures. More specifically, we will show in the following results that there is a clear relationship between declining convective activity and increasing tropospheric ozone above KL-Java in the early months of the year.

The following five figures show MLR monthly trends calculated for our convective proxies shown previously, with the addition of OLR and MERRA-2 PWAT. The trends are presented for four months: March, June, September, and December. **Figure 7** shows the 1998-2022 MLR trends results for the GridSat-B1 11µm $T_b$ over the ESEA region in K per decade. Here,





the black stippling indicates trends that *do not* exceed the 95% confidence interval. The March $T_b$ trends (**Fig 7a**), when KL-Java tropospheric ozone trends are highest, are positive (+3 to +5 K per decade) above KL and Java and throughout the region and exceed the 95% confidence interval. The $T_b$ trends in other months are weak, with the exception of the negative $T_b$ trends south of Java in December (**Fig 7d**). We interpret the March large positive $T_b$ trends as a shift toward less intense and/or less frequent convective activity, which **Figure 3** showed is associated with high tropospheric ozone amounts over KL-Java. The analyses for other datasets will show similar results, as well as similar spatial patterns to the trends shown for $T_b$.

**Figure 8** presents the 1998-2022 NOAA OLR monthly MLR trends in W m$^{-2}$ per decade to supplement the $T_b$ trends and add confidence to our results. The spatial patterns of the OLR trends are nearly identical to $T_b$, with large positive trends (+10 or more W m$^{-2}$ per decade) found in March (**Fig 8a**), and much weaker trends in other months. There are again large negative trends of up to -10 W m$^{-2}$ per decade observed to the south of Java in December (**Fig 8d**). The large positive March OLR trends over ESEA are another indication that convective activity in the early months of the year has waned over the past 25 years. **Figures 7** and **8** show how analogous datasets arrive at this same conclusion.

The 1998-2022 MERRA-2 VP200 MLR trends on **Figure 9** describe how synoptic scale conditions have changed since 1998, and whether large-scale upper tropospheric winds are becoming more or less favorable for convective activity. The March (**Fig 9a**) VP200 trends are highly positive, with per decade changes on the order of the anomalies shown on **Figure 4**. This is a clear sign that upper tropospheric convergence has increased in March, supressing convection as shown on **Figures 7** and **8**, and causing the accumulation of tropospheric ozone and positive ozone trends in the early part of the year. The VP200 trends in other months of the year (**Fig 9b-d**) are weak and do not exceed the 95% confidence interval above KL and Java. These results point to the need to understand potential shifts in the evolution and/or magnitude of the MJO, a primary driver of convective activity over ESEA, over the last 25 years.

The final convective indicator that we examine, MERRA-2 PWAT, is shown on **Figure 10**. PWAT describes the total water vapor content of the atmosphere which generally increases during conditions associated with deep tropical convection. Just as with **Figures 7-9**, the March trends in PWAT point to a large decrease in moisture availability and convective activity, with decreases beyond the 95% confidence interval of -1 to -3 kg m$^{-2}$ over ESEA in March (**Fig 10a**) since 1998. Again, the trends in later months of the year (**Fig 10b-d**) are much weaker with inconclusive results. Taken together, **Figures 7-10** indicate that in the early months of the year there are robust decreases in the strength and frequency of convection, synoptic-scale conditions that have become less favorable for convection, and a drying of the troposphere over KL (yellow box), Java (cyan box), and the ESEA region (black dashed box) as whole.

Over the past ~two decades, emissions controls have substantially decreased the global burden of tropospheric CO. Therefore, we expect to find similar decreases from the 2002/3-2022 AIRS 500 hPa CO MLR trends shown in **Figure 11** (AIRS data are available since September 2002). Indeed, **Figure 11** shows large decreases in tropospheric CO, particularly over China and Australia for most months of the year. Over ESEA, the trends are weaker. However, the positive trends that exceed the 95%





confidence interval near India in March (**Fig 11a**) are notable deviation from this pattern. While not observed directly over our region of interest, these positive CO trends near India in March would act to increase the regional background of ozone, likely contributing to the increases in tropospheric ozone over ESEA. A detailed analysis into whether an increase in biomass burning in the early parts of the year in this region are to blame is beyond the scope of our paper, although the combined effects of

decreasing convection and increasing regional CO in the early part of the year evidently both contribute to the accumulation of tropospheric ozone above KL and Java, leading to the large positive ozone trends observed over the last 25 years.

A summary of *all* monthly trends for the boxed regions shown on previous figures is given in **Figure 12**. Here, the individual data grid points for $T_b$, OLR, VP200, PWAT, and AIRS 500 hPa CO are averaged within the KL (yellow), Java (cyan), and larger region (black) areas, and MLR trends are computed in the same manner as before with the 95% confidence intervals

shown as error bars. **Figure 12a-d** panels show that the datasets we use to convey changes in convective activity peak in March, and for all three regions considered exceed the 95% confidence interval. While AIRS 500 hPa CO (**Fig 12e**) shows decreases in nearly every month of the year for the three regions, the *weakest* trends for all regions coincide with the largest decreases in convection, with even a small positive trend noted over Java (cyan; **Fig 12e**) in April. The lofting and venting of tropospheric pollution (i.e., ozone and CO) has evidently been reduced because of the marked decline in convection in

~February-April, with possible contributions from enhanced biomass burning under the waning convective activity.

## 4 Summary and Conclusions

We used a SOM clustering algorithm and ancillary datasets to describe the relationship between tropospheric ozone, convection, and CO pollution over two tropical ESEA SHADOZ ozonesonde stations, Kuala Lumpur (KL) and Watukosek (Java). The low and high tropospheric ozone clusters are clearly distinguishable by the corresponding anomalies in $T_b$ and

VP200 (indicators of convection) and AIRS tropospheric CO (pollution tracer). The low ozone clusters result from a highly convective environment that is low in background CO due to the lofting of near-surface air and redistribution of pollutants throughout the atmospheric column and downwind. The high ozone clusters are associated with suppressed convective conditions that allow a build-up of pollution (CO), and thus ozone.

Monthly 5-15 km KL-Java ozonesonde profile trends computed with the NASA/GSFC MLR model over the period 1998-2022

show large ozone increases of +4 nmol mol$^{-1}$ or +10 to +15% per decade, primarily confined to the months of February to April. Because the early part of the year is generally convectively active at these stations, we sought to answer whether decreases in convection have led to the observed positive free-tropospheric ozone trends. The MLR trends analysis for the convective indicators $T_b$, OLR, VP200, and PWAT clearly point to the affirmative, and our results show that since 1998 there have been strong decreases in convective activity in the early months of the year. There have been robust increases in average

cloud top brightness temperatures ($T_b$) in conjunction with large OLR and VP200 increases, and decreases in tropospheric moisture content (PWAT) in March.



AIRS 500 hPa CO, while showing negative trends for most months of the year, has increased near India in March, and the overall weakest trends over ESEA are found in March-May with a small increase above Java in April. The combined effect of these trends leads to the accumulation of tropospheric ozone and the +10 to +15% per decade ozone trends found above KL-

Java in February-April. In summary, the seasonal pattern of positive free-tropospheric ozone trends since 1998 over ESEA are largely explained by meteorological, not necessarily chemical, factors. However, possible increases in regional biomass burning (e.g., peat and forest fires) resulting from the reduced convection must be considered. Nonetheless, changing dynamics appear to be a primary driver. Future studies should examine meteorological changes when diagnosing regional tropospheric ozone trends and potential shifts in the timing and spatial patterns of biomass burning and ozone precursor emissions in the

tropics through analysis of datasets such as the Global Fire Emissions Database (GFED; **van der Werf et al., 2017**). Note that all the meteorological data, our monthly sonde data and model output are available for comparison to chemistry-climate models.

Our results demonstrate the utility (or advantages) of monthly resolved trends analysis, particularly because trends for both ozone and dynamical indicators were generally weak outside of February-April. Therefore, we strongly recommend that

monthly or seasonal analyses be employed when seeking to attribute observed ozone trends, particularly for regions where tropospheric ozone is highly meteorologically dependent, and emissions changes are less obvious (e.g., compare emissions reductions and large and predictable seasonal tropospheric ozone variations over North America and Europe), such as ESEA and the KL-Java SHADOZ stations.

The 25-yr trends computed here for KL-Java are similar to those determined in T21 where 22 years of profiles were analyzed.

They also resemble trends that **Gaudel et al. (2020; 2023)** have derived from SHADOZ sondes and IAGOS aircraft profiles measured over Malaysia and Indonesia using the quantile regression (QR) method recommended in **Chang et al. (2022)** and **Chang et al. (2023).** As part of our work with the TOAR-II Harmonization and Evaluation of Ground-based Instruments for Free Tropospheric Ozone Measurements (HEGIFTOM) activity, we have begun to use QR analysis to examine the 25-yr SHADOZ ozone profile record. Preliminary results from QR, annually averaged, resemble those from the MLR and provide

useful complementary information. In general, for SHADOZ stations, including those with smaller trends than KL-Java, the most positive trends from QR over 1998-2022 are strongly represented at the minimum-ozone quantiles (largest increases are found for the lowest ozone amounts). The latter data correspond to the early time of year (February-April) when the convective indicators over the same period are most strongly perturbed over ESEA.

*Data availability* All datasets used in this study are openly and publicly accessible:

- Kuala Lumpur and Watukosek V06 SHADOZ Data: https://tropo.gsfc.nasa.gov/shadoz/Archive.html
- GridSat-B1: https://www.ncei.noaa.gov/data/geostationary-ir-channel-brightness-temperature-gridsat-b1/access/
- NOAA CDR OLR: https://www.ncei.noaa.gov/data/outgoing-longwave-radiation-monthly/access/
- AIRS v7 L3 Daily CO: https://acdisc.gesdisc.eosdis.nasa.gov/opendap/Aqua_AIRS_Level3/AIRS3STM.7.0/



- MERRA-2 M2I3NPASM:
  https://disc.gsfc.nasa.gov/datasets/M2I3NPASM_5.12.4/summary?keywords=%22MERRA-2
- MERRA-2 M2TMNXSLV:
  https://disc.gsfc.nasa.gov/datasets/M2TMNXSLV_5.12.4/summary?keywords=tavgM_2d_slv_Nx
- MODIS FIRMS hotspots (shown as an illustration on Figure 1 only):
  https://firms.modaps.eosdis.nasa.gov/download/

Acknowledgments. We are grateful to NASA's UACO and SAGE-III/ISS programs for support to RMS, AMT, and DEK. We thank NOAA/GML for logistics assistance and support to BJJ and PDC. Many thanks to operators and BRIN and the Malaysia Meteorological Department for outstanding data over many years.


*Author contributions* RMS and AMT conceived of the study. RMS performed the analysis and composed the first draft. DEK processed and archived the SHADOZ ozonesonde data. NK, HKA, and DYR collected the ozonesonde data at and operate the Watukosek SHADOZ station. AD, AFJ, MKS, and NBZ collected the ozonesonde data at and operate the Kuala Lumpur SHADOZ station. BJJ and PDC process and quality assure the data sent from and manage logistics for the Watukosek
SHADOZ station. All authors contributed initial comments during the conception of the study and provided edits to the initial paper draft.

*Competing interests* The authors declare no competing interests.

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

500



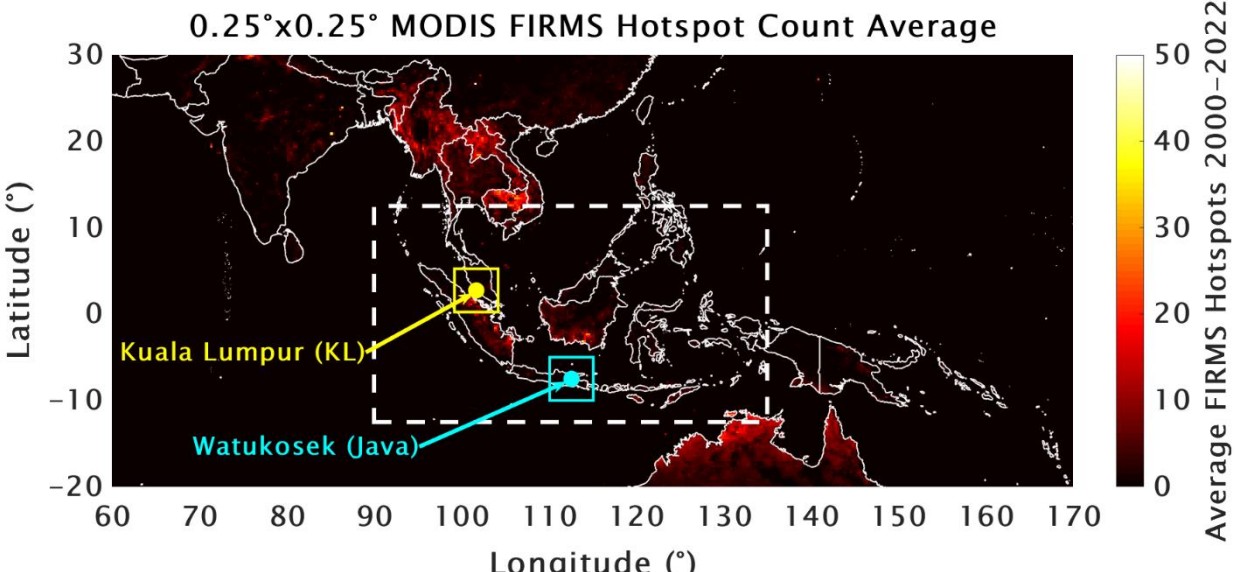

**Figure 1: Map of the study region including average monthly Fire Information for Resource Management System (FIRMS) fire hotspot count from MODIS on the Terra and Aqua satellites for 2000-2022. The boxes indicate specific regions for which monthly trends will later be examined in greater detail. These include Kuala Lumpur (KL; yellow), Watukosek (Java; cyan), and the larger area outlined by the white dashed box.**

505





**Figure 2: The four left panels show ozone mixing ratio SOM output for Kuala Lumpur (a, b) and Watukosek (e, f). The first (low ozone) and fourth (high ozone) nodes are shown for both SHADOZ stations. Grey profiles indicate all members of the respective nodes, with thick black lines showing node mean profiles, and cyan thick and dashed lines showing the average, 20th, and 80th percentile ozone for all profiles from the station. Text in each panel indicates the percent membership and total number of profiles for each node. The right four panels are the percent frequency according to month of the member profiles shown to their left for Kuala Lumpur (c, d) and Watukosek (g, h).**



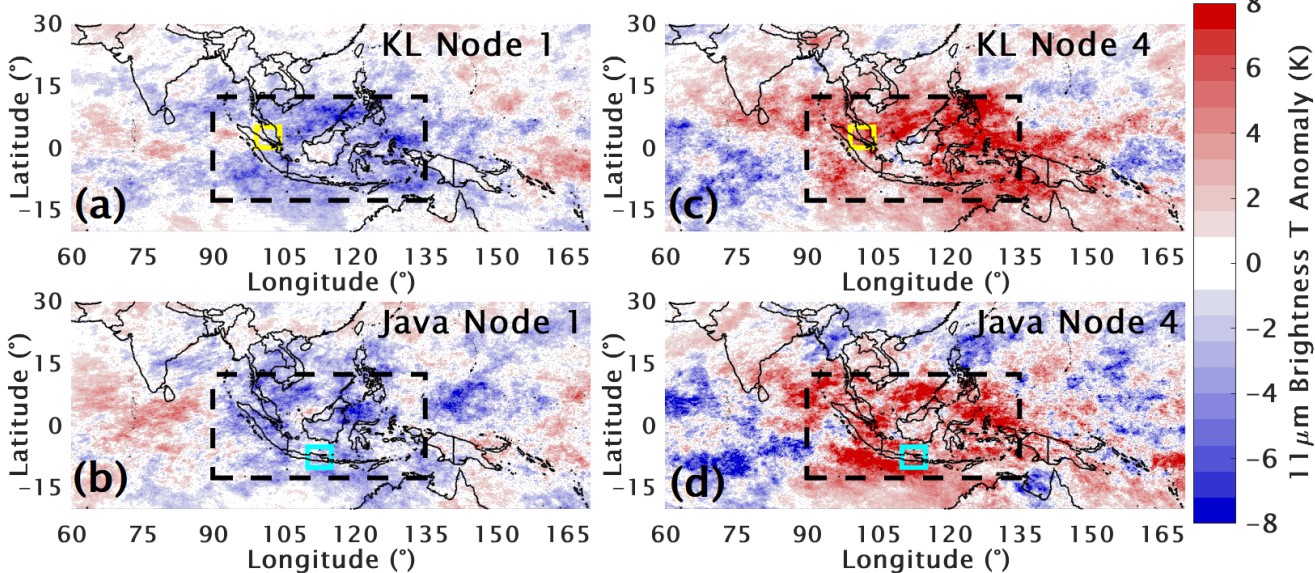

**Figure 3: Composite anomalies from daily climatology (computed for 1991-2020) of GridSat-B1 11 µm brightness temperatures that correspond to the dates of the SOM node member ozonesonde profiles shown on Figure 2. The yellow, cyan, and black dashed boxes highlight the same regions as in Figure 1.**

520



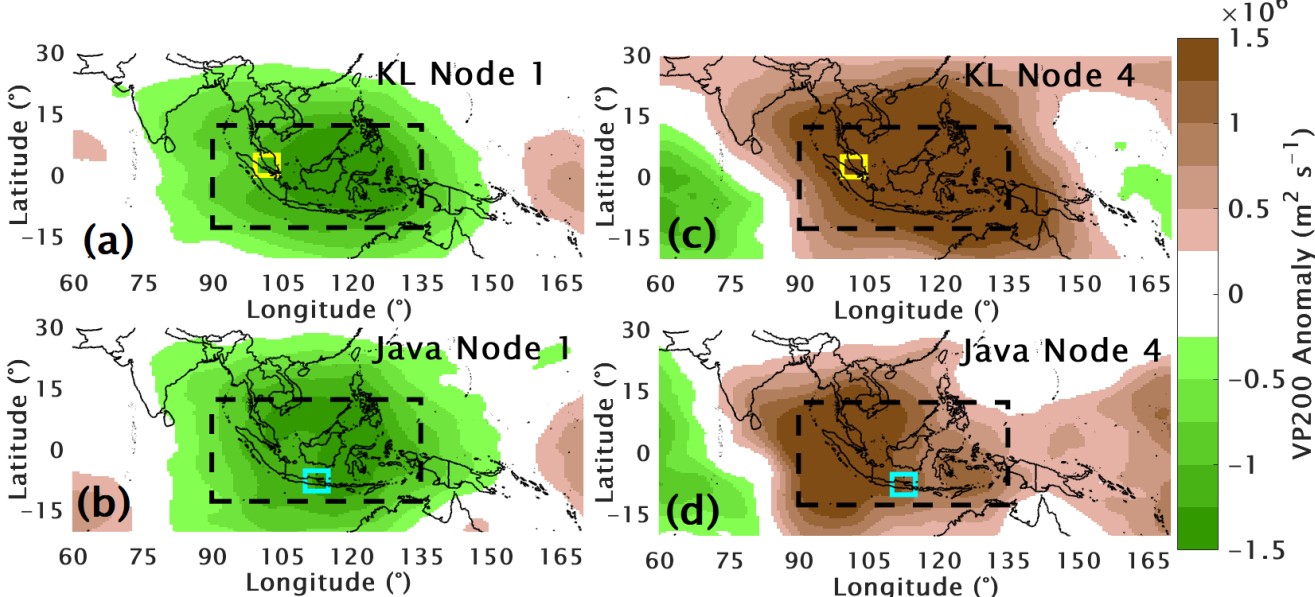

Figure 4: The same as shown in Figure 3, but for anomalies of daily MERRA-2 velocity potential at 200 hPa (VP200).



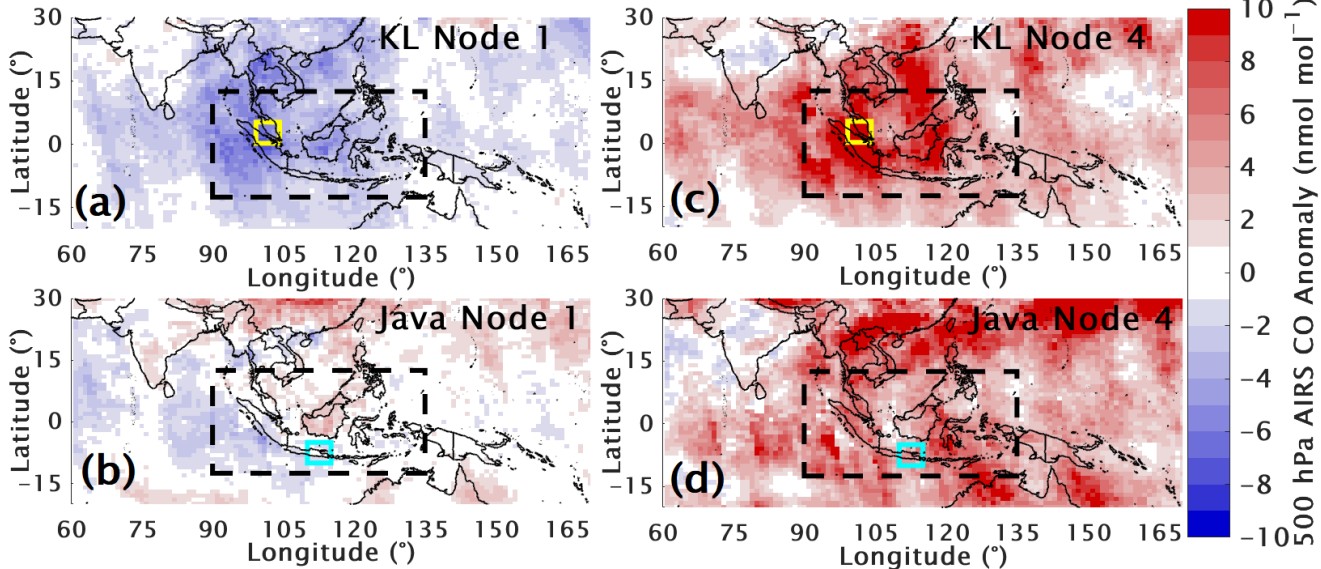

**Figure 5: The same as shown in Figures 3 and 4, but for anomalies of daily AIRS Level 3 carbon monoxide at the 500 hPa level.**



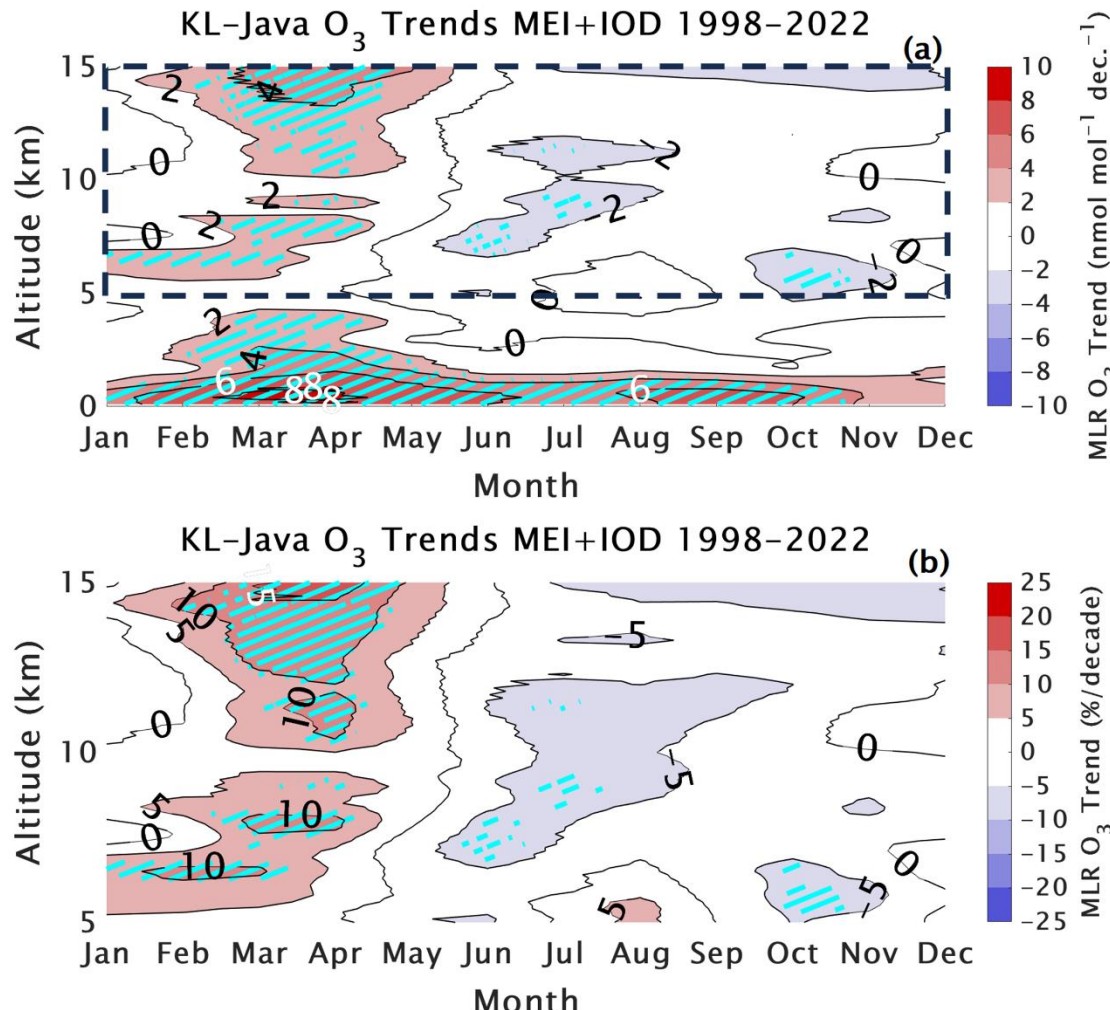

**Figure 6: 1998-2022 monthly multiple linear regression (MLR) model ozone trends for the merged Kuala Lumpur/Watukosek (KL-Java) ozonesonde data in nmol mol$^{-1}$ (a) and percent (b) per decade. Panel (a) includes the lowest 5 km, but our focus is on 5-15 km altitudes as indicated by the black dashed box on (a) and shown in (b). Cyan hatching indicates trends beyond the 95% confidence interval (i.e., historically referred to as *statistically significant*).**



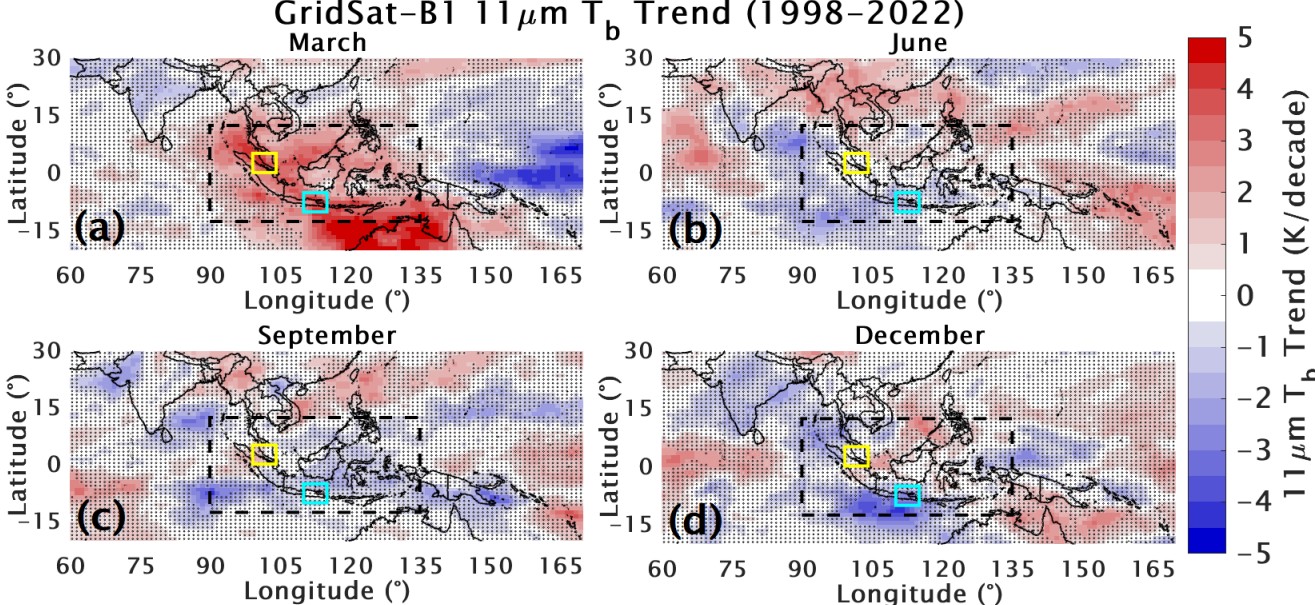

**Figure 7: 1998-2022 MLR trends of GridSat-B1 11 μm brightness temperature in Kelvin per decade for March (a), June (b), September (c), and December (d). The yellow, cyan, and black dashed boxes highlight the same regions as in Figure 1. Stippling indicates trends within the 95% confidence interval (i.e., historically referred to as *statistically insignificant*).**



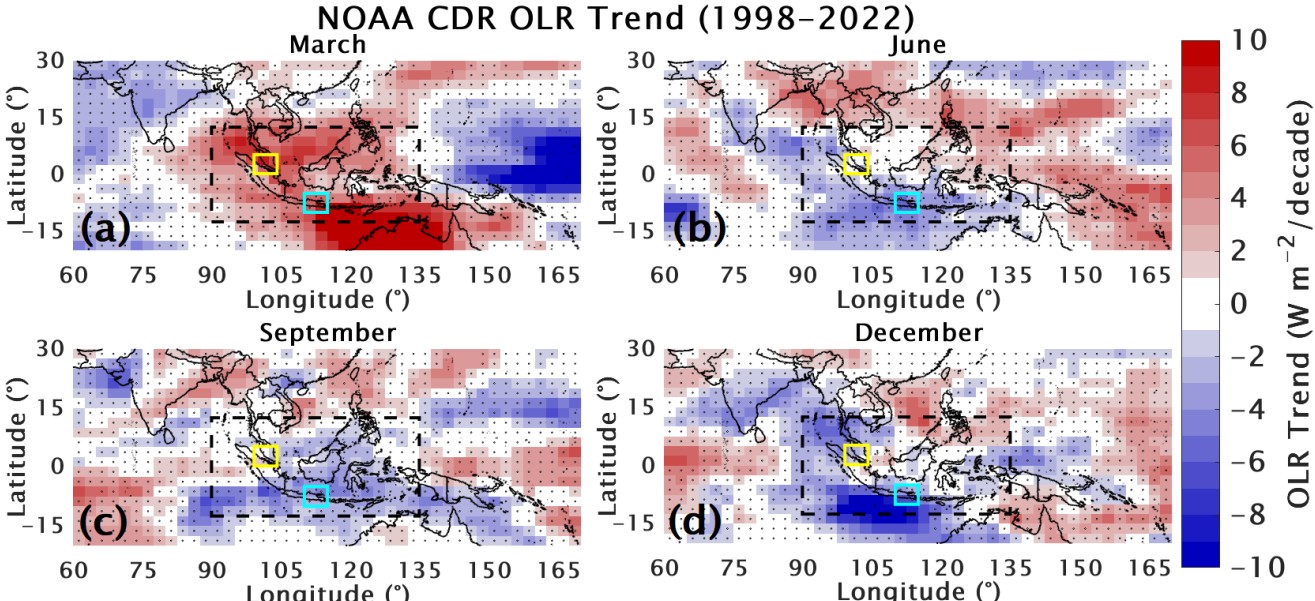

**Figure 8: The same as shown in Figure 7, but for NOAA Climate Data Record (CDR) Outgoing Longwave Radiation (OLR) in Watts per square meter per decade.**



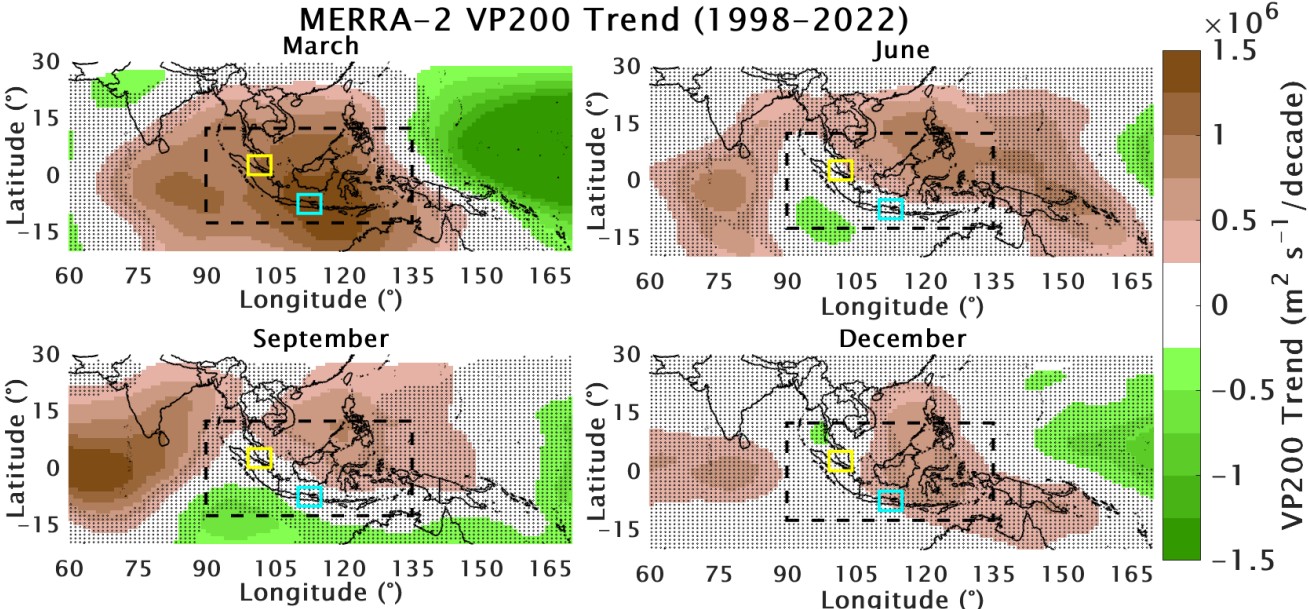

**Figure 9: The same as shown in Figures 7 and 8, but for MERRA-2 velocity potential at 200 hPa (VP200) in meters squared per second per decade.**

545



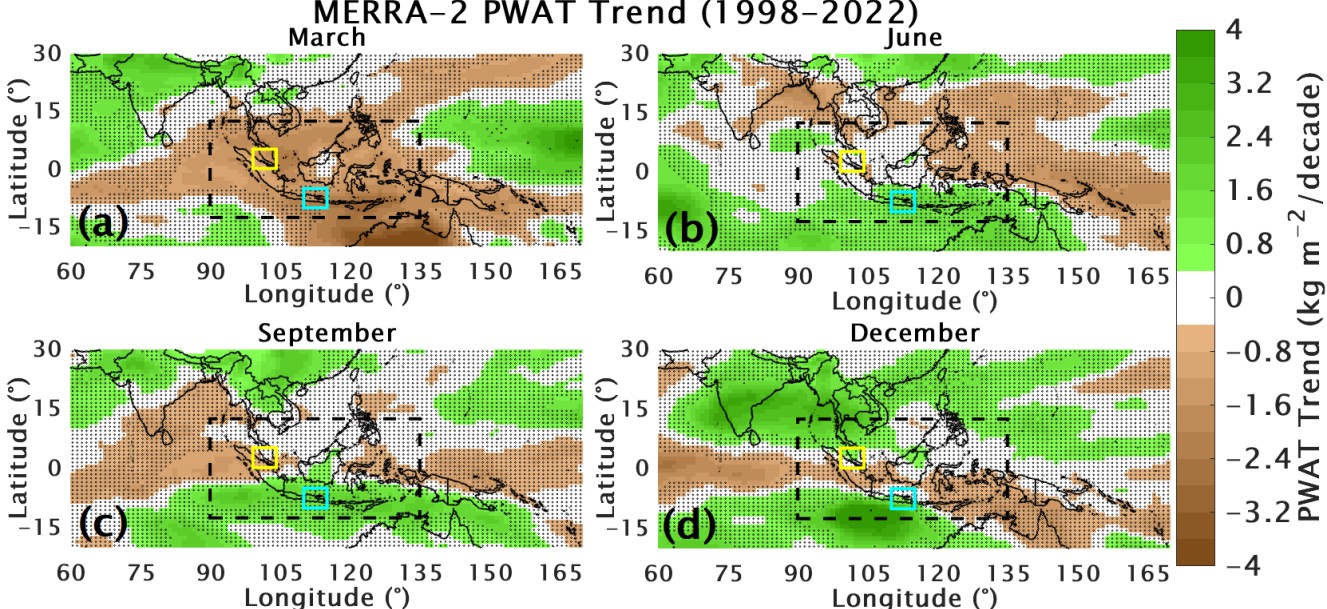

**Figure 10: The same as shown in Figures 7-9, but for MERRA-2 precipitable water (PWAT) in kilograms per square meter per decade.**



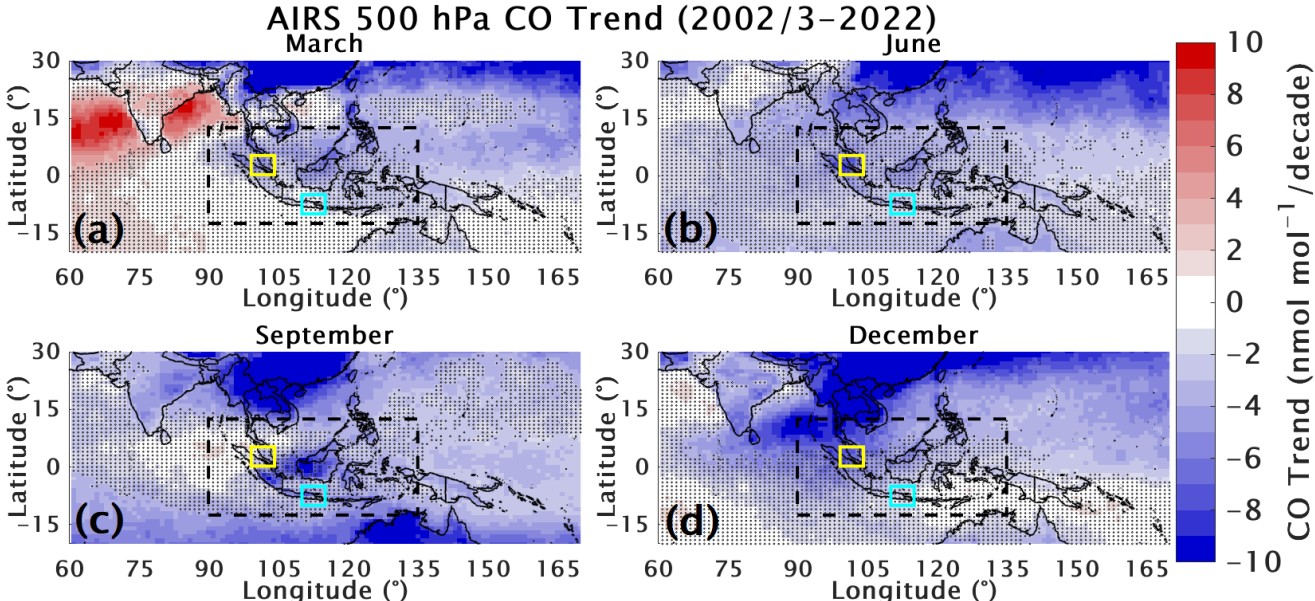

550

**Figure 11: The same as shown in Figures 7-10, but for AIRS CO at 500 hPa in nmol mol$^{-1}$ per decade.**







**Figure 12: Monthly average trends with 95% confidence interval (error bars) for the three regions highlighted on the maps of previous Figures (yellow = Kuala Lumpur; cyan = Watukosek; black = larger region). The monthly trends are presented for GridSat-B1 11 μm T_b (a), NOAA OLR (b), MERRA-2 VP200 (c), MERRA-2 PWAT (d), and AIRS 500 hPa CO (e). All trends are for 1998-2022, except for AIRS CO (e) for which data are only available since September 2002.**