# Peer review of "Dynamical drivers of free-tropospheric ozone increases over equatorial Southeast Asia"

_EGUsphere, 2023_

## Community Comment (CC1)

Comments by Owen R. Cooper (TOAR Scientific Coordinator of the Community Special Issue) on:

**Dynamical drivers of free-tropospheric ozone increases over equatorial Southeast Asia**

Ryan M. Stauffer (corresponding author), Anne M. Thompson, Debra E. Kollonige, Ninong Komala, Habib Khirzin Al-Ghazali, Dian Yudha Risdianto, Ambun Dindang, Ahmad Fairudz bin Jamaluddin, Mohan Kumar Sammathuria, Norazura Binti Zakaria, Bryan J. Johnson, and Patrick D. Cullis

This manuscript was submitted to ACP as part of the TOAR-II Community Special Issue
https://doi.org/10.5194/egusphere-2023-2618

This review is by Owen Cooper (NOAA CSL), TOAR Scientific Coordinator of the TOAR-II Community Special Issue. I, or a member of the TOAR-II Steering Committee, will post comments on all papers submitted to the TOAR-II Community Special Issue, which is an inter-journal special issue accommodating submissions to six Copernicus journals:  ACP (lead journal), AMT, GMD, ESSD, ASCMO and BG. The primary purpose of these reviews is to identify any discrepancies across the TOAR-II submissions, and to allow the author teams time to address the discrepancies.  Additional comments may be included with the reviews.  While O. Cooper and members of the TOAR Steering Committee may post open comments on papers submitted to the TOAR-II Community Special Issue, they are not involved with the decision to accept or reject a paper for publication, which is entirely handled by the journal's editorial team.

This paper is very well written, with a thorough meteorological analysis to demonstrate the impact of seasonal convection patterns on mid- and upper tropospheric ozone above equatorial Southeast Asia.  I recommend some additional text to explain how the current study fits within the context of previous work on the same topic, and to provide some discussion on the relative contributions of changing meteorology and the continuing increase of anthropogenic emissions on the observed increase of ozone above this region.

1) The following statements in the introduction suggest that previous studies have not investigated the impact of climate variability and seasonal cycles on ozone trends and variability:  "the possible effects of dynamics and climate change have been given little consideration." and "Seasonally or monthly resolved analyses are less common (e.g., Chang et al., 2023; Section 3.4)".
There is a very large body or work that addresses the impact of climate change on ozone, summarized by several review papers and IPCC AR6 (Jacob and Winner, 2009; Fiore et al., 2012; Fiore et al., 2015; von Schneidemesser et al., 2015; Szopa et al., 2021).  Many studies have examined how trends vary by season or with climate variability (such as ENSO), and it is now standard procedure for modeling studies to quantify the impact of meteorological variability on ozone trends (Columbi et al., 2023; Cooper, M.J. et al., 2013; Li S. et al., 2023; Lin et al. 2014,2015,2017; Rowlinson et al., 2019; Wang et al. 2022a; Wang et al 2022b; Xue et al. 2020).  To provide a broader context for the submitted paper it would be helpful to point out the new aspects of this study and how they build on earlier work.

2)  Detailed budget studies on the drivers of ozone trends across the tropics began in the mid-1990s with the development of global scale three-dimensional atmospheric chemistry models. The earliest studies indicate that increasing anthropogenic emissions are the primary cause of increasing tropical ozone (Levy et al., 1997; Roelofs et al., 1997).  Since that time models and emissions inventories have

continued to improve and successive generations of models (Szopa et al., 2021; Skeie et al, 2020; Griffiths et al., 2021; Liu et al., 2022) have attributed the observed ozone increases in the tropics to anthropogenic and biomass burning emissions, with anthropogenic emissions continuing to increase in the region of SE Asia (Li, M. et al., 2023). Two recent model studies explored the relative contributions of changing emissions and meteorological variability across SE Asia and concluded that rising emissions are driving the ozone increase (Wang et al., 2022b; Li. S. et al., 2023). The submitted paper does not address the impact of rising emissions on the observed ozone variability in the ozonesonde record, and some discussion is needed to quantify the relative contributions of dynamical changes and rising ozone precursors.

3) Several papers in the literature have discussed the impact of ozone sampling frequency and the challenges of detecting trends (Prinn 1988; Chang et al., 2020), or calculating accurate monthly or seasonal mean ozone values (Logan, 1999, Saunois et al., 2012). These earlier studies focused on northern mid-latitudes and a new study submitted to the TOAR-II Community Special Issue addresses this challenge at a tropical location (Chang et el., 2024). Some discussion is needed regarding the ozonesonde sample size and the confidence in the reported trends.

**References**

Chang, K.-L., Cooper, O. R., Gaudel, A., Petropavlovskikh, I., Effertz, P., Morris, G., and McDonald, B. C.: Technical note: Challenges of detecting free tropospheric ozone trends in a sparsely sampled environment, EGUsphere [preprint], https://doi.org/10.5194/egusphere-2023-2739, 2024.

Colombi, N. K., Jacob, D. J., Yang, L. H., Zhai, S., Shah, V., Grange, S. K., Yantosca, R. M., Kim, S., and Liao, H.: Why is ozone in South Korea and the Seoul metropolitan area so high and increasing?, Atmos. Chem. Phys., 23, 4031–4044, https://doi.org/10.5194/acp-23-4031-2023, 2023

Cooper, M. J., R. V. Martin, N. J. Livesey, D. A. Degenstein, K. A. Walker, Analysis of satellite remote sensing observations of low ozone events in the tropical upper troposphere and links with convection. Geophys. Res. Lett. 40, 3761–3765 (2013)

Fiore, A.M., Naik, V., Spracklen, D.V., Steiner, A., Unger, N., Prather, M., Bergmann, D., Cameron-Smith, P.J., Cionni, I., Collins, W.J. and Dalsøren, S., 2012. Global air quality and climate. Chemical Society Reviews, 41(19), pp.6663-6683.

Fiore, A.M., Naik, V. and Leibensperger, E.M., 2015. Air quality and climate connections. Journal of the Air & Waste Management Association, 65(6), pp.645-685.

Griffiths, P. T., Murray, L. T., Zeng, G., Shin, Y. M., Abraham, N. L., Archibald, A. T., Deushi, M., Emmons, L. K., Galbally, I. E., Hassler, B., Horowitz, L. W., Keeble, J., Liu, J., Moeini, O., Naik, V., O'Connor, F. M., Oshima, N., Tarasick, D., Tilmes, S., Turnock, S. T., Wild, O., Young, P. J., and Zanis, P.: Tropospheric ozone in CMIP6 simulations, Atmos. Chem. Phys., 21, 4187–4218, https://doi.org/10.5194/acp-21-4187-2021, 2021

Jacob, D.J. and Winner, D.A., 2009. Effect of climate change on air quality. Atmospheric environment, 43(1), pp.51-63.

Levy, H., Kasibhatla, P.S., Moxim, W.J., Klonecki, A.A., Hirsch, A.I., Oltmans, S.J. and Chameides, W.L., 1997. The global impact of human activity on tropospheric ozone. Geophysical Research Letters, 24(7), pp.791-794

Li, M., Kurokawa, J., Zhang, Q., Woo, J.-H., Morikawa, T., Chatani, S., Lu, Z., Song, Y., Geng, G., Hu, H., Kim, J., Cooper, O. R., and McDonald, B. C.: MIXv2: a long-term mosaic emission inventory for Asia (2010–2017), EGUsphere [preprint], https://doi.org/10.5194/egusphere-2023-2283, 2023

Li, S., Yang Yang, Hailong Wang, Pengwei Li, Ke Li, Lili Ren, Pinya Wang, Baojie Li, Yuhao Mao, Hong Liao (2023), Rapid increase in tropospheric ozone over Southeast Asia attributed to changes in precursor emission source regions and sectors, Atmos. Environ., https://doi.org/10.1016/j.atmosenv.2023.119776

Lin, M., Horowitz, L.W., Oltmans, S.J., Fiore, A.M. and Fan, S., 2014. Tropospheric ozone trends at Mauna Loa Observatory tied to decadal climate variability. Nature Geoscience, 7(2), pp.136-143

Lin, M., Fiore, A.M., Horowitz, L.W., Langford, A.O., Oltmans, S.J., Tarasick, D. and Rieder, H.E., 2015. Climate variability modulates western US ozone air quality in spring via deep stratospheric intrusions. Nature communications, 6(1), p.7105.

Lin, M., et al. (2017), US surface ozone trends and extremes from 1980 to 2014: quantifying the roles of rising Asian emissions, domestic controls, wildfires, and climate, Atmos. Chem. Phys., 17, 2943–2970, 2017, www.atmos-chem-phys.net/17/2943/2017/doi:10.5194/acp-17-2943-2017

Liu, J., Strode, S. A., Liang, Q., Oman, L. D., Colarco, P. R., Fleming, E. L., et al. (2022). Change in tropospheric ozone in the recent decades and its contribution to global total ozone. *Journal of Geophysical Research: Atmospheres*, *127*, e2022JD037170. https://doi.org/10.1029/2022JD037170

Logan, J. A.: An analysis of ozonesonde data for the troposphere: Recommendations for testing 3-D models and development of a gridded climatology for tropospheric ozone, Journal of Geophysical Research: Atmospheres, 104, 16 115–16 149, 1999

Prinn, R.G., 1988. Toward an improved global network for determination of tropospheric ozone climatology and trends. Journal of Atmospheric Chemistry, 6, pp.281-298.

Roelofs, G.J., Lelieveld, J. and van Dorland, R., 1997. A three-dimensional chemistry/general circulation model simulation of anthropogenically derived ozone in the troposphere and its radiative climate forcing. *Journal of Geophysical Research: Atmospheres*, *102*(D19), pp.23389-23401.

Rowlinson, M.J. et al., 2019: Impact of El Niño-Southern Oscillation on the interannual variability of methane and tropospheric ozone. Atmospheric Chemistry and Physics, 19(13), 8669–8686, doi:10.5194/acp-19-8669-2019

Saunois, M., Emmons, L., Lamarque, J.-F., Tilmes, S., Wespes, C., Thouret, V., and Schultz, M.: Impact of sampling frequency in the analysis of tropospheric ozone observations, Atmospheric Chemistry and Physics, 12, 6757–6773, https://doi.org/10.5194/acp-12-6757-2012, 2012.

Skeie, R.B., Myhre, G., Hodnebrog, Ø., Cameron-Smith, P.J., Deushi, M., Hegglin, M.I., Horowitz, L.W., Kramer, R.J., Michou, M., Mills, M.J. and Olivié, D.J., 2020. Historical total ozone radiative forcing derived

from CMIP6 simulations. *Npj Climate and Atmospheric Science*, *3*(1), p.32, https://www.nature.com/articles/s41612-020-00131-0

Szopa, S., V. Naik, B. Adhikary, P. Artaxo, T. Berntsen, W.D. Collins, S. Fuzzi, L. Gallardo, A. Kiendler-Scharr, Z. Klimont, H. Liao, N. Unger, and P. Zanis, 2021: Short-Lived Climate Forcers. In Climate Change 2021: The Physical Science Basis. Contribution of Working Group I to the Sixth Assessment Report of the Intergovernmental Panel on Climate Change [Masson-Delmotte, V., P. Zhai, A. Pirani, S.L. Connors, C. Péan, S. Berger, N. Caud, Y. Chen, L. Goldfarb, M.I. Gomis, M. Huang, K. Leitzell, E. Lonnoy, J.B.R. Matthews, T.K. Maycock, T. Waterfield, O. Yelekçi, R. Yu, and B. Zhou (eds.)]. Cambridge University Press, Cambridge, United Kingdom and New York, NY, USA, pp. 817–922, doi:10.1017/9781009157896.008

von Schneidemesser, E. et al., 2015: Chemistry and the Linkages between Air Quality and Climate Change. Chemical Reviews, 115(10), 3856–3897, doi:10.1021/acs.chemrev.5b00089

Wang, H., Lu, X., Jacob, D. J., Cooper, O. R., Chang, K.-L., Li, K., Gao, M., Liu, Y., Sheng, B., Wu, K., Wu, T., Zhang, J., Sauvage, B., Nédélec, P., Blot, R., and Fan, S. (2022a), Global tropospheric ozone trends, attributions, and radiative impacts in 1995–2017: an integrated analysis using aircraft (IAGOS) observations, ozonesonde, and multi-decadal chemical model simulations, Atmos. Chem. Phys., 22, 13753–13782, https://doi.org/10.5194/acp-22-13753-2022

Wang, X., et al. (2022b), Rapidly Changing Emissions Drove Substantial Surface and Tropospheric Ozone Increases Over Southeast Asia, Geophysical Research Letters, 49, e2022GL100223. https://doi.org/10.1029/2022GL100223

Xue, L., A. Ding, O. Cooper, X. Huang, W. Wang, D. Zhou, Z. Wu, A. McClure-Begley, I. Petropavlovskikh, M. O. Andreae, C. Fu (2020), ENSO and Southeast Asian biomass burning modulate subtropical trans-Pacific ozone transport, National Science Review, nwaa132, https://doi.org/10.1093/nsr/nwaa132

---

## Author Comment (AC1)

*Note: Author responses are in blue, italicized text.*

This is a well written paper with reasonable conclusions. In particular, the SOM analysis seems to do a good job of selecting more strongly convectively influenced versus subsidence profiles. And the discussion around the importance of diagnosing the seasonal variation in the ozone trend is very useful to say. My main comment is that the paper seems longer than necessary due to using so many convective proxies, rather than simply just using rain itself (more in comment below). The technical standard of the paper was quite high, and I don't have any significant comments to make on these aspects of the paper.

*We thank the Reviewer for the thoughtful comments. They inspired us to generate more analyses in the form of Figures for Reviewers that bolster our results indicating the tropospheric ozone/convection trends relationship even further. As the Reviewer noted that they thought the paper was a bit long anyway, we decided not to include these Reviewer Figures in the manuscript.*

The SHADOZ sondes also contain RH, and it would have been interesting to do a trend analysis of that quantity also, since it is also linked to convection. Could be issues with the data of course, especially in the upper troposphere.

*This is a good suggestion. We converted the SHADOZ radiosonde RH measurements into water vapor mixing ratio and calculated monthly 5-15 km MLR trends in the exact same way as with ozone. This is shown in Figure R1. There has been a significant moistening of the lower troposphere in most months after April, but the only significant decrease in water vapor mixing ratio is in fact found in March. This corresponds with the PWAT analysis (Figure 10) contained in the manuscript. We agree that the radiosonde RH (and the derived water vapor mixing ratio) above ~12 or 13 km should always be taken with a grain of salt.*

[Figure]

*Figure R1. 1998-2022 Monthly MLR trends of water vapor mixing ratio (in grams per kilogram per decade) derived from SHADOZ radiosonde (coupled to the*

*ozonesondes) profiles. As in the manuscript, cyan hatching denotes trends that exceed the 95% confidence interval.*

The authors might have considered rain itself as a more direct proxy for convection than water column, cloud brightness temperature, OLR, velocity potential, etc. There may have been a bit of "over analysis" here in using all these proxies instead of going straight to rain. I think the paper would be more useful if a direct connection between ozone and rain could be established, since rainfall is more directly relevant. Presumably, some version of the TRMM dataset (3 hourly) would be appropriate. I realize rain is a more "noisy" variable, but it is also more interesting.

*In the paper we analyzed multiple observational and assimilated datasets to show that several independent physical phenomena arrived at the same conclusion of waning convection in ~Feb-Apr from 1998-2022, leading to the observed positive free-tropospheric (5-15km) ozone trends. Because convective precipitation is a mesoscale feature often influenced by land and topography, while VP200/MJO, ENSO etc., act on synoptic or even larger scales, it's better to look at those regional metrics rather than more localized (rainfall) conditions to diagnose FT ozone variability and trends. Nonetheless, here we show an analysis similar to Figures in the manuscript for Global Precipitation Climatology Project (GPCP; https://psl.noaa.gov/data/gridded/data.gpcp.html; Figure R2) and MERRA-2 convective precipitation (Figure R3). As with the GridSat-B1 product, GPCP is Climate Data Record (CDR) quality. Indeed, the GPCP data are a bit noisier than the proxies that we analyzed, but both datasets also arrive at the same conclusion. Note that in both Figure R2 and R3 that the precipitation trend spatial patterns, especially in March, correspond to those in Figures 7-10 in the manuscript. We have chosen to retain our analyses and Figures in the paper, but hope you find these Figures for Reviewers insightful.*

[Figure]

*Figure R2. 1998-2022 MLR GPCP (2.5°x2.5°) total precipitation trends in mm per day per decade for March, June, September, and December. As in the manuscript, stippling shows trends within the 95% confidence interval bounds (i.e., "insignificant").*

[Figure]

*Figure R3. 1998-2022 MLR MERRA-2 (1.0°x1.0°) convective precipitation trends in kg per square meter per second per decade for March, June, September, and December. As in the manuscript, stippling shows trends within the 95% confidence interval bounds (i.e., "insignificant").*

It might be borderline appropriate to call the ozone trend using 22 years of data a trend instead of some form of longer term variance. In this context, it also might be worth mentioning, or even showing, the absolute ozone trends (with MJO, ENSO variability included), since there could be changes in the fraction of the total rain/ozone variance that are part of these oscillations, so that there may not be an absolute trend in these quantities.

*We assume you mean 25 years (1998-2022) of data, not 22. We include an ENSO proxy (in the form of MEIv2) in the MLR trend model. When ENSO is accounted for in the model, and despite the last three years of our analysis ending in a "triple dip" La Nina which lowers tropospheric ozone in this region, we still find strongly positive ~Feb-Apr free-tropospheric (5-15 km) ozone trends. You bring up a good point about whether these trends will continue in the future. Clearly there has been a change in MJO (as given by the VP200 proxy) and thus FT ozone in the early part of the year in this region over the last 25 years. Whether the VP200/OLR/$T_B$ trends calculated here are the result of changes or shifts in the spatial patterns, seasonality, strength, etc. of MJO, ENSO, and their interactions with each other is beyond the scope of our study, but they are interesting questions nonetheless and should be examined in future efforts.*

*We show here (Figure R4) the relationships between MEIv2 and ozone, and VP200 anomalies (computed for the black dash boxed region on previous Figures) and ozone, further indicating that any trend or change in VP200 anomalies in particular will result in tropospheric ozone changes. The relationship between these quantities and ozone is stronger for the 5-15 km layer (top of Figure R4) than the surface-5 km layer (bottom of Figure R4). The fact that there is still a weak relationship between surface-5 km ozone and VP200 anomalies may result in the correspondence between near-surface and FT ozone trends in Feb-Apr (Figure 6, and as you note below).*

[Figure]

*Figure R4. Scatterplots of 5 to 15 km (top row) and surface to 5 km (bottom row) SHADOZ monthly partial column ozone anomalies corresponding to MEIv2 (left column) and VP200 anomaly values (right column). VP200 anomalies are computed for the black dash boxed region shown on Figures in the manuscript and above.*

The positive ozone trends near the surface in the top panel of Fig. 6 from June onward look like they are confined to the boundary layer. Comparison with the stability could confirm. This would make more physical sense than saying "below 700 hPa". Could simply then say that, for whatever reason, BL increases in ozone are not being communicated to the free troposphere (maybe since BL increases are a local response to emissions increases).

*Another good suggestion. We computed 1998-2022 monthly MLR trends (Figure R5) for the vertical gradient of potential temperature from the SHADOZ radiosondes to investigate changes in boundary layer stability and possible links to the ozone trends shown on Figure 6a. Significant negative trends in dθ/dz (less stability) below 2 km are found in ~Mar-Jul. This does not perfectly align with the trends in Figure 6a, but may help somewhat explain why positive ozone trends between about 2 and 5 km are found only in ~Feb-Apr. We also point out the increasing trend in stability at 14-15 km in March in*

*Figure R5, another indication of decreasing favorability for convection. The opposite is true in ~Jul-Oct, when negative ozone trends are found at these altitudes in Figure 6.*

*Thank you for the suggestion on wording. We have added this sentence to Section 3.2 the paper: "However, these near-surface trends are not being communicated to the FT for most of the year. This indicates that surface ozone trends may be primarily driven by emissions changes, while FT trends, as we will show, are the result of changes to convective activity. The trends appear to be somewhat independent of each other in the two different segments of the profile."*

[Figure]

*Figure R5. 1998-2022 Monthly MLR trends of the vertical potential temperature gradient (in Kelvin per kilometer per decade) derived from SHADOZ radiosonde (coupled to the ozonesondes) profiles. As in the manuscript, cyan hatching denotes trends that exceed the 95% confidence interval.*

---

## Author Comment (AC2)

*Note: Author responses are in blue, italicized text.*

This is a well-motivated and well-focused manuscript that investigates trends in tropospheric ozone from ozonesonde observations at two stations in equatorial Southeast Asia and their association with concomitant changes in observed convection (as deduced from multiple remotely sensed cloud metrics and environmental proxies). This was one of the most polished papers I've received in review and I enjoyed reading it from start to finish. I applaud the authors for their crafting of an excellent narrative with appropriate concision, clear goals, and unexaggerated interpretations. The figure quality was also superb and each well justified and discussed. Congratulations on a wonderful effort that I believe is ready to be published as is, with only a single technical correction - "trends" on line 89 should be "trend".

*Thank you for the encouraging words on our manuscript. We have made the technical correction listed above. You may wish to also peruse our responses to the other two sets of comments, which contain further evidence of the so-called "ozone/convection tuning knob" and that trends in convection are a primary driver of the 25-year positive free-tropospheric ozone trends above Equatorial Southeast Asia.*

---

## Author Comment (AC3)

*Note: Author responses are in blue, italicized text.*

Comments by Owen R. Cooper (TOAR Scientific Coordinator of the Community Special Issue) on: Dynamical drivers of free-tropospheric ozone increases over equatorial Southeast Asia Ryan M. Stauffer (corresponding author), Anne M. Thompson, Debra E. Kollonige, Ninong Komala, Habib Khirzin Al-Ghazali, Dian Yudha Risdianto, Ambun Dindang, Ahmad Fairudz bin Jamaluddin, Mohan Kumar Sammathuria, Norazura Binti Zakaria, Bryan J. Johnson, and Patrick D. Cullis

This manuscript was submitted to ACP as part of the TOAR-II Community Special Issue https://doi.org/10.5194/egusphere-2023-2618 Preprint. Discussion started: November 4, 2023; discussion closes January 20, 2024 This review is by Owen Cooper (NOAA CSL), TOAR Scientific Coordinator of the TOAR-II Community Special Issue. I, or a member of the TOAR-II Steering Committee, will post comments on all papers submitted to the TOAR-II Community Special Issue, which is an inter-journal special issue accommodating submissions to six Copernicus journals: ACP (lead journal), AMT, GMD, ESSD, ASCMO and BG. The primary purpose of these reviews is to identify any discrepancies across the TOAR-II submissions, and to allow the author teams time to address the discrepancies. Additional comments may be included with the reviews. While O. Cooper and members of the TOAR Steering Committee may post open comments on papers submitted to the TOAR-II Community Special Issue, they are not involved with the decision to accept or reject a paper for publication, which is entirely handled by the journal's editorial team.

This paper is very well written, with a thorough meteorological analysis to demonstrate the impact of seasonal convection patterns on mid- and upper tropospheric ozone above equatorial Southeast Asia. I recommend some additional text to explain how the current study fits within the context of previous work on the same topic, and to provide some discussion on the relative contributions of changing meteorology and the continuing increase of anthropogenic emissions on the observed increase of ozone above this region.

*We thank O. Cooper for helpful contextual comments, additional references, and pointers on topics to address in the manuscript text.*

1) The following statements in the introduction suggest that previous studies have not investigated the impact of climate variability and seasonal cycles on ozone trends and variability: "the possible effects of dynamics and climate change have been given little consideration." and "Seasonally or monthly resolved analyses are less common (e.g., Chang et al., 2023; Section 3.4)". There is a very large body or work that addresses the impact of climate change on ozone, summarized by several review papers and IPCC AR6 (Jacob and Winner, 2009; Fiore et al., 2012; Fiore et al., 2015; von Schneidemesser et al., 2015; Szopa et al., 2021). Many studies have examined how trends vary by season or with climate variability (such as ENSO), and it is now standard procedure for modeling studies to quantify the impact of meteorological variability on ozone trends (Columbi et al., 2023; Cooper, M.J. et al., 2013; Li S. et al., 2023; Lin et al. 2014,2015,2017; Rowlinson et al., 2019; Wang et al. 2022a; Wang et al 2022b; Xue et al. 2020). To provide a broader context for the submitted paper it would be helpful to point out the new aspects of this study and how they build on earlier work.

*We have removed this statement in Section 1: "However, the possible effects of dynamics and climate change have been given little consideration. This is somewhat surprising…".*

*We have also made this edit to Section 1: "There is a tendency to report tropospheric ozone trends using a single (annually averaged) value over some period of interest. Seasonally or monthly resolved analyses are less common (e.g., Chang et al., 2023; Section 3.4)." We strongly believe that these statements are still true. For example, as prominently displayed in the BAMS State of the Climate Report published each year.*

*The novelty of this work is the unambiguous result linking free-tropospheric (5-15 km) ozone trends to trends in convective parameters from multiple observational datasets. This was achieved using the highest-quality, homogenized, 100 meter resolution vertical ozonesonde profiles. The ozone trend computation by itself is not so original, but to our knowledge no other study has shown such a conclusive link between ozone and convective trends in this region. Our hope is that this motivates modelers to re-examine their simulations of free-tropospheric ozone and model convective parameters. Indeed, this is something we are exploring in preparation for the July 2024 Quadrennial Ozone Symposium. These mechanisms should also be explored elsewhere in the tropics and globally.*

2) Detailed budget studies on the drivers of ozone trends across the tropics began in the mid-1990s with the development of global scale three-dimensional atmospheric chemistry models. The earliest studies indicate that increasing anthropogenic emissions are the primary cause of increasing tropical ozone (Levy et al., 1997; Roelofs et al., 1997). Since that time models and emissions inventories have continued to improve and successive generations of models (Szopa et al., 2021; Skeie et al, 2020; Griffiths et al., 2021; Liu et al., 2022) have attributed the observed ozone increases in the tropics to anthropogenic and biomass burning emissions, with anthropogenic emissions continuing to increase in the region of SE Asia (Li, M. et al., 2023). Two recent model studies explored the relative contributions of changing emissions and meteorological variability across SE Asia and concluded that rising emissions are driving the ozone increase (Wang et al., 2022b; Li. S. et al., 2023). The submitted paper does not address the impact of rising emissions on the observed ozone variability in the ozonesonde record, and some discussion is needed to quantify the relative contributions of dynamical changes and rising ozone precursors.

*There is little doubt that the near-surface ozone trends of 4+ nmol mol$^{-1}$ per decade are the result of local and regional emissions increases. We have now noted that in Section 3.2. Again, that segment of the profile is not the primary focus of our paper, especially because the free and upper troposphere are where ozone climate radiative forcing impacts are the greatest. However, Reviewer 1 brought up a good point about the lack of "communication" between the near-surface and free-tropospheric trends. A more detailed examination of the relationship between ENSO, VP200, and the near-surface and FT segments of the profile suggests that the trends in the two different layers are likely somewhat independent of each other. That is, the FT trends are driven by changes to ENSO, MJO (VP200), and convection, while the near-surface is likely more sensitive to emissions changes, and less sensitive to ENSO and MJO.*

*From our response to Reviewer 1: "We show here (Figure R4) the relationships between MEIv2 and ozone, and VP200 anomalies (computed for the black dash boxed region on previous Figures) and ozone, further indicating that any trend or change in VP200 anomalies in particular will result in tropospheric ozone changes. The relationship between these quantities and ozone is stronger for the 5-15 km layer (top of Figure R4) than the surface-5 km layer (bottom of Figure R4). The fact that there is still a weak relationship between surface-5 km ozone and VP200 anomalies may result in the correspondence between near-surface and FT ozone trends in Feb-Apr (Figure 6, and as you note below)." The data unequivocally show how strong the relationship is between FT ozone and VP200, and that the SHADOZ monthly means are more than sufficient to describe the convection/FT ozone interactions.*

3) Several papers in the literature have discussed the impact of ozone sampling frequency and the challenges of detecting trends (Prinn 1988; Chang et al., 2020), or calculating accurate monthly or seasonal mean ozone values (Logan, 1999, Saunois et al., 2012). These earlier studies focused on northern mid-latitudes and a new study submitted to the TOAR-II Community Special Issue addresses this challenge at a tropical location (Chang et el., 2024). Some discussion is needed regarding the ozonesonde sample size and the confidence in the reported trends.

*The Chang et al., (2024) paper is a nice resource for understanding sampling and ozone trends calculated for sub-tropical and higher latitude locations. Our analyses (as with Thompson et al., 2021) are restricted to stations in the deep tropics within ~10 degrees of the Equator. Variability induced by STE, for example, below 15 km will be minimal. Achieving higher confidence in our calculated trends in addition to providing attribution is precisely why we analyzed so many ancillary datasets, all of them with daily (OLR, AIRS CO) or sub-daily (GridSat-B1, MERRA-2) resolution. In response to Reviewer 1, we also examined precipitation data – further confirmation of the other independent parameters. They each arrive at the same conclusion, that ESEA convection has waned in ~Feb-Apr over the last 25 years, which matches the patterns in free-tropospheric ozone trends at the two SHADOZ stations. Our ozone trend results for the ESEA stations are essentially confirmed by examining the larger picture of MJO, ENSO, etc. variability. Again, we do not dismiss the impact of emissions increases on the strongly positive near-surface ozone trends.*

*Figure R4 (provided in the Response to Reviewer 1) shows the relationship between monthly averages of tropospheric ozone and MEIv2 (ENSO), and VP200 anomalies (MJO), and that the SHADOZ sampling is sufficient to capture the expected covariance in these metrics.*

[Figure]

*Figure R4. Scatterplots of 5 to 15 km (top row) and surface to 5 km (bottom row) partial column ozone anomalies corresponding to MEIv2 (left column) and VP200 anomaly values (right column). VP200 anomalies are computed for the black dash boxed region shown on Figures in the manuscript and above.*

*Thank you for providing the references listed below.*

References

Chang, K.-L., Cooper, O. R., Gaudel, A., Petropavlovskikh, I., Effertz, P., Morris, G., and McDonald, B. C.: Technical note: Challenges of detecting free tropospheric ozone trends in a sparsely sampled environment, EGUsphere [preprint], https://doi.org/10.5194/egusphere-2023-2739, 2024.

Colombi, N. K., Jacob, D. J., Yang, L. H., Zhai, S., Shah, V., Grange, S. K., Yantosca, R. M., Kim, S., and Liao, H.: Why is ozone in South Korea and the Seoul metropolitan area so high and increasing?, Atmos. Chem. Phys., 23, 4031–4044, https://doi.org/10.5194/acp-23-4031-2023, 2023

Cooper, M. J., R. V. Martin, N. J. Livesey, D. A. Degenstein, K. A. Walker, Analysis of satellite remote sensing observations of low ozone events in the tropical upper troposphere and links with convection. Geophys. Res. Lett. 40, 3761–3765 (2013)

Fiore, A.M., Naik, V., Spracklen, D.V., Steiner, A., Unger, N., Prather, M., Bergmann, D., Cameron-Smith, P.J., Cionni, I., Collins, W.J. and Dalsøren, S., 2012. Global air quality and climate. Chemical Society Reviews, 41(19), pp.6663-6683.

Fiore, A.M., Naik, V. and Leibensperger, E.M., 2015. Air quality and climate connections. Journal of the Air & Waste Management Association, 65(6), pp.645-685.

Griffiths, P. T., Murray, L. T., Zeng, G., Shin, Y. M., Abraham, N. L., Archibald, A. T., Deushi, M., Emmons, L. K., Galbally, I. E., Hassler, B., Horowitz, L. W., Keeble, J., Liu, J., Moeini, O., Naik,

V., O'Connor, F. M., Oshima, N., Tarasick, D., Tilmes, S., Turnock, S. T., Wild, O., Young, P. J., and Zanis, P.: Tropospheric ozone in CMIP6 simulations, Atmos. Chem. Phys., 21, 4187–4218, https://doi.org/10.5194/acp-21-4187- 2021, 2021

Jacob, D.J. and Winner, D.A., 2009. Effect of climate change on air quality. Atmospheric environment, 43(1), pp.51-63. Levy, H., Kasibhatla, P.S., Moxim, W.J., Klonecki, A.A., Hirsch, A.I., Oltmans, S.J. and Chameides, W.L., 1997. The global impact of human activity on tropospheric ozone. Geophysical Research Letters, 24(7), pp.791-794

Li, M., Kurokawa, J., Zhang, Q., Woo, J.-H., Morikawa, T., Chatani, S., Lu, Z., Song, Y., Geng, G., Hu, H., Kim, J., Cooper, O. R., and McDonald, B. C.: MIXv2: a long-term mosaic emission inventory for Asia (2010–2017), EGUsphere [preprint], https://doi.org/10.5194/egusphere-2023-2283, 2023

Li, S., Yang Yang, Hailong Wang, Pengwei Li, Ke Li, Lili Ren, Pinya Wang, Baojie Li, Yuhao Mao, Hong Liao (2023), Rapid increase in tropospheric ozone over Southeast Asia attributed to changes in precursor emission source regions and sectors, Atmos. Environ., https://doi.org/10.1016/j.atmosenv.2023.119776

Lin, M., Horowitz, L.W., Oltmans, S.J., Fiore, A.M. and Fan, S., 2014. Tropospheric ozone trends at Mauna Loa Observatory tied to decadal climate variability. Nature Geoscience, 7(2), pp.136-143

Lin, M., Fiore, A.M., Horowitz, L.W., Langford, A.O., Oltmans, S.J., Tarasick, D. and Rieder, H.E., 2015. Climate variability modulates western US ozone air quality in spring via deep stratospheric intrusions. Nature communications, 6(1), p.7105.

Lin, M., et al. (2017), US surface ozone trends and extremes from 1980 to 2014: quantifying the roles of rising Asian emissions, domestic controls, wildfires, and climate, Atmos. Chem. Phys., 17, 2943–2970, 2017, www.atmos-chem-phys.net/17/2943/2017/doi:10.5194/acp-17-2943-2017

Liu, J., Strode, S. A., Liang, Q., Oman, L. D., Colarco, P. R., Fleming, E. L., et al. (2022). Change in tropospheric ozone in the recent decades and its contribution to global total ozone. Journal of Geophysical Research: Atmospheres, 127, e2022JD037170. https://doi.org/10.1029/2022JD037170

Logan, J. A.: An analysis of ozonesonde data for the troposphere: Recommendations for testing 3-D models and development of a gridded climatology for tropospheric ozone, Journal of Geophysical Research: Atmospheres, 104, 16 115–16 149, 1999

Prinn, R.G., 1988. Toward an improved global network for determination of tropospheric ozone climatology and trends. Journal of Atmospheric Chemistry, 6, pp.281-298. Roelofs, G.J., Lelieveld, J. and van Dorland, R., 1997. A three-dimensional chemistry/general circulation model simulation of anthropogenically derived ozone in the troposphere and its radiative climate forcing. Journal of Geophysical Research: Atmospheres, 102(D19), pp.23389-23401.

Rowlinson, M.J. et al., 2019: Impact of El Niño-Southern Oscillation on the interannual variability of methane and tropospheric ozone. Atmospheric Chemistry and Physics, 19(13), 8669–8686, doi:10.5194/acp-19-8669-2019

Saunois, M., Emmons, L., Lamarque, J.-F., Tilmes, S., Wespes, C., Thouret, V., and Schultz, M.: Impact of sampling frequency in the analysis of tropospheric ozone observations, Atmospheric Chemistry and Physics, 12, 6757–6773, https://doi.org/10.5194/acp-12-6757-2012, 2012.

Skeie, R.B., Myhre, G., Hodnebrog, Ø., Cameron-Smith, P.J., Deushi, M., Hegglin, M.I., Horowitz, L.W., Kramer, R.J., Michou, M., Mills, M.J. and Olivié, D.J., 2020. Historical total ozone radiative forcing derived from CMIP6 simulations. Npj Climate and Atmospheric Science, 3(1), p.32, https://www.nature.com/articles/s41612-020-00131-0

Szopa, S., V. Naik, B. Adhikary, P. Artaxo, T. Berntsen, W.D. Collins, S. Fuzzi, L. Gallardo, A. Kiendler Scharr, Z. Klimont, H. Liao, N. Unger, and P. Zanis, 2021: Short-Lived Climate Forcers. In Climate Change 2021: The Physical Science Basis. Contribution of Working Group I to the Sixth Assessment Report of the Intergovernmental Panel on Climate Change [Masson-Delmotte, V., P. Zhai, A. Pirani, S.L. Connors, C. Péan, S. Berger, N. Caud, Y. Chen, L. Goldfarb, M.I. Gomis, M. Huang, K. Leitzell, E. Lonnoy, J.B.R. Matthews, T.K. Maycock, T. Waterfield, O. Yelekçi, R. Yu, and B. Zhou (eds.)]. Cambridge University Press, Cambridge, United Kingdom and New York, NY, USA, pp. 817–922, doi:10.1017/9781009157896.008

von Schneidemesser, E. et al., 2015: Chemistry and the Linkages between Air Quality and Climate Change. Chemical Reviews, 115(10), 3856–3897, doi:10.1021/acs.chemrev.5b00089

Wang, H., Lu, X., Jacob, D. J., Cooper, O. R., Chang, K.-L., Li, K., Gao, M., Liu, Y., Sheng, B., Wu, K., Wu, T., Zhang, J., Sauvage, B., Nédélec, P., Blot, R., and Fan, S. (2022a), Global tropospheric ozone trends, attributions, and radiative impacts in 1995–2017: an integrated analysis using aircraft (IAGOS) observations, ozonesonde, and multi-decadal chemical model simulations, Atmos. Chem. Phys., 22, 13753–13782, https://doi.org/10.5194/acp-22-13753-2022

Wang, X., et al. (2022b), Rapidly Changing Emissions Drove Substantial Surface and Tropospheric Ozone Increases Over Southeast Asia, Geophysical Research Letters, 49, e2022GL100223. https://doi.org/10.1029/2022GL100223 Xue, L., A. Ding, O. Cooper, X. Huang,

W. Wang, D. Zhou, Z. Wu, A. McClure-Begley, I. Petropavlovskikh, M. O. Andreae, C. Fu (2020), ENSO and Southeast Asian biomass burning modulate subtropical transPacific ozone transport, National Science Review, nwaa132, https://doi.org/10.1093/nsr/nwaa132